# The Feasibility of Make My Day—A Randomized Controlled Pilot Trial of a Stroke Prevention Program in Primary Healthcare

**DOI:** 10.3390/ijerph20196828

**Published:** 2023-09-25

**Authors:** Emelie Mälstam, Eric Asaba, Elisabet Åkesson, Susanne Guidetti, Ann-Helen Patomella

**Affiliations:** 1Department of Neurobiology, Care Sciences and Society, Division of Occupational Therapy, Karolinska Institutet, 141-52 Stockholm, Sweden; eric.asaba@ki.se (E.A.); susanne.guidetti@ki.se (S.G.); ann-helen.patomella@ki.se (A.-H.P.); 2Department of Occupational and Public Health Sciences, Faculty of Health and Occupational Studies, University of Gävle, 801-76 Gävle, Sweden; 3Unit for Research, Development, and Education, Stockholm’s Sjukhem Foundation, 112-19 Stockholm, Sweden; elisabet.akesson@stockholmssjukhem.se; 4Department of Neurobiology, Care Sciences and Society, Division of Neurogeriatrics, Karolinska Institutet, 141-52 Stockholm, Sweden; 5Women’s Health and Allied Health Professionals Theme Medical Unit Occupational Therapy and Physiotherapy, Karolinska University Hospital, 171-76 Stockholm, Sweden

**Keywords:** engaging occupation, mHealth, occupation-focused, occupational literacy, stroke prevention

## Abstract

Incorporating and sustaining engaging everyday activities (EEAs) in everyday life holds potential for improving health and wellbeing; thus, there is reason to explore EEAs as a behavioral change technique in stroke prevention. The aim of this study was to evaluate the feasibility of the stroke prevention program Make My Day (MMD) for people with moderate-to-high risk for stroke in a primary healthcare setting, where EEAs are utilized to promote healthy activity patterns. A randomized controlled pilot trial was designed to evaluate the feasibility of MMD. Twenty-nine persons at risk for stroke were recruited and randomized into either an intervention group (*n* = 14) receiving MMD or a control group (*n* = 15) receiving brief health advice and support with goal setting. The results suggest that MMD is feasible, with timely recruitment, overall high response rates and study completion, and sensitivity to change in key outcome measures. Moreover, the results demonstrate that the application of EEAs can be useful for promoting behavioral change in stroke prevention. Recommendations for improvements for a full-scale trial include recruiting a relevant sample, using reliability- and validity-tested outcome measures, and implementing strategies to limit missing data.

## 1. Introduction

This study evaluates the feasibility of a prevention program focusing on lifestyle factors and engaging everyday activities (EEAs) to address noncommunicable diseases (NCDs) such as stroke [1]. The concept of lifestyle is often associated with preferences in different activities (e.g., food choices, exercise, smoking) [2,3]. However, the idea that lifestyle is about personal choices and preferences has been challenged conceptually [4]. Lifestyle can also involve individual and collective choices, behaviors, and everyday activities situated in a sociocultural context [5,6]. Health interventions that target changes in lifestyle-related risk factors must thus consider and balance the situatedness of people’s lifestyles.

Stroke and other NCDs can be the objects of lifestyle-focused interventions [7], since these diseases share similar risk factors—many of which are considered to be modifiable (e.g., unhealthy diet, physical inactivity, tobacco and alcohol consumption, having a high body mass index (BMI), and hypertension) [8]. Multiple risk factors are related to stroke, and multi-morbidity [9] is a major challenge in working with this type of behavioral change [10]. Incorporating healthy lifestyle habits into everyday life is strongly associated with reduced stroke risk [11,12,13]. However, it is reasonable to assume that lifestyle-focused stroke prevention should promote meaningful and purposeful activities and habits situated in everyday life, preferentially with people who personally matter to the individual in question. Since the late 1990s, interventions with such a focus in combination with a lifestyle redesign have shown positive results in participation, physical functioning, and various dimensions of quality of life in well elderly people [14,15,16,17,18,19], as well as positive clinical outcomes in persons at risk for cardiovascular disease and those with already-established cardiovascular disease or diabetes [20,21]. On this basis, we argue that, in the context of stroke prevention, EEAs hold greater potential than other activities for being incorporated and sustained in everyday life.

In this study, we use EEAs to refer to socioculturally situated activities that the individual perceives as highly meaningful, valuable, purposeful, and providing positive feelings [22]. EEAs can bring an intense sense of participation and losing track of time; they can be frequently integrated into everyday life and performed regularly. EEAs can also hold value for others (e.g., family, friends, or society at large), creating motivation for engagement in other activities in life [6,22]. They are not confined to a specific arena in life; rather, EEAs can be hobbies, physical exercise, work, and household activities. Even sedentary leisure activities have been described as EEAs (e.g., watching television or online gaming), although a large amount of these activities in everyday life can be seen as negative in terms of individual health from a stroke risk perspective. While not all EEAs can be considered as health-promoting [23], EEAs hold potential to promote a healthy lifestyle when redesigned with the purpose of increasing health and well-being or decreasing the risk of ill health.

### The Make My Day Intervention

In Make My Day (MMD), EEAs are utilized as a means and goal for changing and sustaining a healthy lifestyle; they are facilitated through an individual lifestyle analysis, physical group sessions, and an mHealth app (mobile health technology) for individualized self-monitoring (Figure 1). With support from an occupational therapist and the Canadian Occupational Performance Measure (COPM) [24], persons at risk for stroke formulate personally relevant and meaningful lifestyle goals related to EEAs. In contrast to other studies using COPM to focus on problem areas in the performance of everyday activities, COPM was utilized in this study to support individuals in identifying and prioritizing areas in their lifestyle that are important for their health and well-being, and in setting goals to incorporate more EEAs into their everyday life.

MMD has previously been modeled and tested for feasibility in a multiple case study among persons at risk for stroke who also had a transient ischemic attack (TIA) [25], as recommended by the Medical Research Council (MRC) guidelines [26,27]. The results from the case study were positive in terms of participant acceptability and improvements in reaching lifestyle goals; however, MMD was tested outside of primary healthcare services in the case study, making it important to pilot MMD in a healthcare setting, where it is intended to be implemented. The need to develop and evaluate stroke prevention for people with different levels of risk for stroke has also been emphasized [9,28]. Evaluating MMD among persons at risk for stroke but without earlier TIA incidence or stroke is therefore suitable. A pilot study evaluation using a randomized controlled trial (RCT) can further improve the design of MMD and trial procedures in preparation for a full-scale trial [29,30].

The aim in the present study was to evaluate the feasibility of the stroke prevention program MMD for persons with moderate-to-high risk for stroke in a Swedish primary healthcare setting, with a focus on the following:(1)the success of recruitment and retention procedures;(2)the acceptability and suitability of the instruments and response rates;(3)adherence to the MMD pilot trial and prevention program;(4)sensitivity to change in the outcome measures.

## 2. Materials and Methods

### 2.1. Study Design

A two-armed randomized controlled assessor-blinded parallel pilot trial was designed to evaluate the feasibility of MMD [1] in a Swedish primary healthcare setting. The focus of the trial was on recruitment and retention procedures, the acceptability and suitability of instruments (online or paper surveys), response rates, participant adherence to the study and intervention program, contextual factors influencing data collection, and the sensitivity to change of the outcome measures. The trial followed the Consolidated Standards of Reporting Trials (CONSORT) extension on randomized pilot and feasibility trials for the reporting of the study [31] and was registered as a clinical trial at Clinicaltrials.gov (NCT03730701).

### 2.2. Recruitment of Persons at Risk for Stroke

During 2018–2019, five primary healthcare sites were invited to co-deliver MMD together with the research group; two sites participated in the study. The enrollment of study participants was scheduled between June 2019 and June 2020 and was conducted through digital advertising, local newspapers, and flyers. The inclusion criteria for participation were (a) age of 45–75 years; (b) three or more stroke risk factors scored as moderate-to-high risk on a stroke risk scorecard; (c) access to a smartphone or tablet; (d) expressed motivation for lifestyle change; and (e) understanding of the Swedish language. The exclusion criteria were (a) history of a previous stroke or TIA diagnosis; and/or (b) expressed ongoing drug abuse.

Given the study aim, a prospective sample size was not required. Instead, the enrollment aimed for a sample size large enough to provide useful information for the study [29,30]. At each recruited site, 15 participants were aimed for, to make it possible to conduct group sessions while achieving good group dynamics. A dropout rate of 20% was viewed as acceptable.

### 2.3. Randomization

Following baseline assessments, the participants were allocated to the intervention group (IG) or control group (CG). The allocation was carried out using a randomized blinded block randomization (2 + 2) by a researcher uninvolved in data collection or MMD (see page 3 in the previously published study protocol for the randomization procedure) [1]. The code key for the allocation of the study participants to the two respective groups was locked in a department safe in a sealed envelope, which was not opened until all 12-month follow-ups were completed.

### 2.4. Intervention Group: Make My Day

Persons in the IG participated in MMD (Figure 1). During the first individual session, the participants worked one-on-one with a registered occupational therapist who asked about the participant’s personal background and medical history, and screened for stroke risk, stroke risk literacy, current life situation, and activity pattern. In addition, COPM [24] was used to define and prioritize areas in the participants’ lifestyle that they considered important to their health and well-being. The occupational therapist used COPM to facilitate the articulation of personally relevant and meaningful lifestyle goals related to EEAs during a week, such as performing more Zumba, calling a friend, or planning for grocery shopping to facilitate healthy meals or snacks. The goals were added into the MMD mHealth app and continuously aimed for by the participants. The MMD mHealth app was an integrated part of the MMD program, as it augmented the physical onsite group sessions with digital self-monitoring of lifestyle goals, EEAs, physical activity, nutrition, stress, and/or tobacco and alcohol consumption. Push messages and personalized results from individual registrations were also sent. During the physical group sessions, which were held in parallel with the participants’ engagement with the app, the participants interacted with health professionals, including an occupational therapist, physiotherapist, and dietitian. Sessions with health professionals were used to reflect and learn about EEAs and healthy activity patterns, and to try out health-promoting activities (e.g., walk-and-talk, food preparation, yoga, and other forms of exercise) together in the group.

### 2.5. Control Group

The persons randomized into the CG were provided with an equivalent lifestyle analysis and goal setting by an occupational therapist during the baseline assessments; the therapist also provided brief advice on lifestyle-focused prevention of stroke, which was personalized according to the participants’ lifestyle analysis. All participants in the study (IG and CG) that were identified as having a medically uncontrolled risk factor (e.g., high blood pressure) were referred to a general practitioner (GP) and/or specialist nurse in primary healthcare.

### 2.6. Data Collection

Feasibility data and outcome measurements were collected at several time points throughout the study period during 2018–2020 (Figure 2). The interventionist researchers (n = 2) and research assistant (n = 1) kept logbooks of the participants’ recruitment, randomization procedures, response rates, adherence (i.e., number of participants at each group session and reasons for non-attendance), and follow-up rates, as well as wrote reflective field notes about the study context. Data were collected by three blinded researchers (the first and last authors and a research assistant) following a program manual. Blinding was safeguarded by ensuring that the assessors did not deliver the intervention to the participants they assessed and by informing the participants they met not to reveal their allocation. Aside from some exceptions (due to illness), the same assessor followed up with the same participants from baseline to the 12-month follow-up.

Participants were provided with the option of answering self-administered paper questionnaires or online surveys. Paper-based surveys were returned in a prepaid/pre-addressed envelope, while online surveys were accessed through a link sent by e-mail. Three reminders to respond to the questionnaires were sent by e-mail. Data collected in person included background information, stroke risk, stroke risk literacy, COPM, blood pressure, weight and height, and physical performance. Assessments were conducted at the primary healthcare clinics or at university facilities.

Due to the Corona Virus (COVID) 19 pandemic, the 12-month follow-up was modified to align with physical distancing regulations. All assessments that were originally supposed to be performed in person were performed over telephone, except for measurements of weight, blood pressure, and physical performance, which were performed (if acceptable to the participants) outside of their home or workplace, in what we termed a “mobile lab”. In this way, the participants were not required to use public transportation, and the assessors could maintain physical distance. Through this mobile lab, the study participants were provided with equipment and oral and written instructions with visual aids for measuring their weight and blood pressure, while the physical performance test was performed by the assessor.

The acceptability of delivering or participating in MMD and the mechanisms of impact, such as the reach of primary healthcare clinics, fidelity, and dose of MMD, have been explored in a separate study [in manuscript]. Similarly, the usability of the MMD [submitted manuscript] and experiences of the MMD mHealth app in relation to changing lifestyle habits [32] are discussed in other articles.

### 2.7. Outcome Measurements

Several outcome measurements that were planned for application in the full-scale trial of MMD were tested for their acceptability, suitability, and sensitivity to change in this pilot study. An evaluation of acceptability and suitability was conducted by examining participants’ experiences of filling out the instrument and by counting the number of answered surveys. Sensitivity to change was evaluated by analyzing the instruments’ ability to detect changes during the study period (Table 1). Outcome measurements included overall stroke risk [33]; individual stroke risk factors (i.e., blood pressure, BMI, physical activity, dietary habits, alcohol and tobacco habits); physical performance [34]; activity performance and satisfaction [24]; participation in health-promoting activities; healthy activity patterns [35]; activity balance [36]; stroke risk literacy [37]; and quality of life [38,39].

#### 2.7.1. Overall Stroke Risk

The Stroke Risk Score Card (SRSC) from the National Stroke Association in America was used to assess participants’ overall stroke risk. The 2018 version of the SRSC [33] includes eight risk factors. Each risk factor was given a score in one of three columns (high risk, caution, and low risk). Each column was summed to a total count, resulting in an overall stroke risk with information on what the person needed to address moving forward. The overall stroke risks were as follows: high risk ≥ 3: Ask about stroke prevention right away; caution 4–6: A good start. Work on reducing risk; low risk 6–8: You’re doing very well at controlling stroke risk! To test the overall reduction in stroke risk, participants’ scores were transformed into a dichotomous variable. If participants changed their overall stroke risk either from high risk to caution or from caution to low risk, this was classified as an improvement (overall stroke risk reduction); if the stroke risk changed from low risk to caution, from caution to high risk, or did not change at all, this was classified as non-improvement (no overall stroke risk reduction).

#### 2.7.2. Individual Risk Factors for Stroke

The following individual risk factors for stroke were examined: blood pressure, BMI, and different lifestyle habits. Systolic and diastolic blood pressure was measured in mmHg with an Omron M4 Intelli IT monitor after at least 10 min of rest sitting down. Weight was measured with a Beurer BF 180 personal weight scale; height was reported by the participants in centimeters; and BMI was calculated by dividing weight by height **×** height (kg/m^2^).

Using the Swedish National Board of Health and Welfare Questionnaires on Lifestyle Habits [40], the participants self-rated their lifestyle habits during the last few months. Tobacco consumption was scored on a 7-point scale (1 = No, I have never smoked to 7 = Yes, I smoke more than 10 cigarettes every day). Alcohol consumption was scored on a 5-point scale (1 = None at all or less than 1 glass each week to 5 = more than 14 glasses each week). Physical exercise and activity were scored on a 6-point scale (1 = No, almost never to 7 = More than 300 min every week). Dietary habits (vegetable, fruit, or snack consumption) were scored on a 4-point scale (1 = Once a week or less, 2= a few times a week, 3 = once every day, 4 = twice every day or more).

#### 2.7.3. Physical Performance

Physical performance was measured using the 6 min walk test (6MWT) [34], which is a simple cardiopulmonary functional testing modality that reflects usual everyday activity and exercise performance. The primary test outcome was the final distance measured in meters that was walked in 6 min, where 400–700 m is a normal walking distance for adults. The test was conducted according to the manual, with appropriate equipment (e.g., a Beurer PO 30 pulse oximeter, Borg scale, and stopwatch).

#### 2.7.4. Activity Performance

COPM assesses performance and satisfaction of lifestyle goals related to EEAs from the individual’s perspective using a 1–10 scale. Each participant was asked 1) to rate the performance of the specified activity using a 1–10 scale, and 2) to score his or her satisfaction with that performance using the same scale. Two summative scores for performance and satisfaction were created by adding the scores from the chosen activities together and then dividing by the number of activities chosen. The summative scores were divided by the number of rated activities to provide an instrument-specific score that was used for comparisons. A change in the mean score of two points was viewed as clinically meaningful [24].

#### 2.7.5. Healthy Activity Patterns

The Daily Experience of Pleasure, Productivity, and Restoration (PRP) Profile [35] is a time use diary designed to collect objective and subjective information about people’s everyday activities within a 24 h time frame. This instrument considers both time use and different experiences associated with well-being (i.e., pleasure, productivity, and restoration). In our study, an additional health experience category was added to assess participants’ experience of their health when performing different activities, after discussion with and approval from the PRP developer, Karen Atler. In the modified PRP profile, the participants were asked to keep a record of the activities they completed during a day, along with information on when, where, and with whom they performed each activity. All participants were asked to record their activities from three random days (24 h) during the course of one week. Each activity was scored using a 7-point response scale (1–7) of the four experience-based categories (i.e., pleasure, productivity, restoration, and health). We extracted data on time spent on different activities and the scoring of health experience. The average value of the number of hours spent on activities per registered day between 7 am and 10 pm was calculated.

#### 2.7.6. Participation in Health-Promoting Activities

To study participation in health-promoting activities (PHPA) in everyday life, a new instrument was created and tested. The aim was to capture the dimension of engagement in everyday activity—a dimension that was perceived to be missing in the other questionnaires, but relevant for the intervention. The questionnaire contained 10 items on perceptions of health and other experiences in everyday activities, such as meaning, joy, pleasure, creativity, happiness, and positive challenges. Each item was scored with a 5-point scale on the frequency of participation in different activities during the last month (1 = No, I usually do not participate in this kind of activity, 2 = Sometimes, less often than once a week, 3 = Often, 1–2 times a week, 4 = Regularly, three or more times a week, 5 = Daily).

#### 2.7.7. Activity Balance

The Occupational Balance Questionnaire (OBQ) [36] was used to assess participants’ activity balance in relation to their current situation and daily life. Thirteen affirmations in OBQ were scored using a 4-point response scale (0–3) according to the level of agreement with each affirmation (0 = Completely disagree to 3 = Completely agree). A total score was summed up with a range from 0 to 39, where a greater score indicated greater activity balance in everyday life.

#### 2.7.8. Stroke Risk Literacy

Stroke risk literacy was explored using three relevant items from a questionnaire focusing on knowledge and awareness of stroke and risk perceptions [37]. The participants were asked (1) to identify risk factors for stroke, (2) to identify which risk factors could be modified by lifestyle habits, and (3) to score their own risk for stroke on a 10-point visual analogue scale. The number of risk factors and lifestyle influences was counted and compared with the outcome assessment to look for changes in stroke risk literacy.

#### 2.7.9. Quality of Life

Quality of life was assessed with two different instruments, EQ-5D [38] and LiSat-11 [39]. EQ-5D is a short questionnaire with five dimensions, answered with a severity scale from 1 to 3 (EQ-5D-3L) and the EQ-Visual Analogue Scale (EQ-VAS). EQ-VAS is a vertical health barometer from 0 to 100 measuring the experience of health that day, with endpoints labeled “The best health you can imagine” (100) and “The worst health you can imagine” (0). LiSat-11 measures perceived life satisfaction. The first item in the assessment, LiSat-11-1 (Perceived overall life satisfaction), was included and scored on a 6-point scale (1 = Very dissatisfied to 6 = Very satisfied) [39].

### 2.8. Data Analysis

The feasibility of the design regarding recruitment, randomization, and retention rates was compared with the study protocol [1] and program manual, to determine the potential for realizing a full-scale RCT. Descriptive statistics were used to examine the distribution of data, and to present the recruitment, retention, response rates, demographics, and characteristics of the persons at risk for stroke.

A paired sample *t*-test was used for normally distributed data and continuous variables to compare outcomes at different time points within the respective IG and CG, while an independent sample *t*-test was applied when comparing the outcomes between the IG and CG; these results were displayed with the mean and standard deviation (SD). A Wilcoxon signed rank test was used to compare non-normally distributed data and ordinal data within the IG and CG, while a Mann–Whitney U-test was applied for comparison between the IG and CG; these results were displayed with the median and inter-quartile range (IQR). Due to a small sample, Fisher’s exact test was used to determine significant associations.

A logistic regression analysis was performed with a bilateral stroke risk outcome (improved overall stroke risk or no improvement) with the independent variable of treatment group (IG vs. CG) to study the odds ratio (OR) and confidence intervals (CIs). Controlling of covariates (i.e., age, gender, country of birth, living situation, education, income, and work situation) was achieved by analyzing one covariate at a time.

The level of statistical significance was set at *p* ≤ 0.05 and the confidence level was set at 95%. SPSS V.22.0 and SAS 9.4 were used to analyze the data. Clinically meaningful changes were presented for COPM [24], EQ-VAS [38], and blood pressure [40,41].

### 2.9. Ethical Considerations

Ethical approval was obtained from the Swedish Ethical Review Authority Stockholm Ethical Committee, Sweden (No. 2015/834-31, 2016/2203-32, 2019-01444, 2020-03822). Enrollment in the pilot trial was based on written and oral informed consent obtained at several time points, as the data were collected over time (2019–2020). The assessors were registered health professionals and researchers, with experience in both clinical practice and research. If study participants exhibited unexpected symptoms, they were referred to relevant professionals in primary healthcare. The trial was conducted according to Quality Assurance of Clinical Research, which all investigators had completed training in prior to the start of the trial.

## 3. Results

The feasibility of MMD is presented in four sections: (1) participant recruitment and baseline characteristics, (2) the acceptability and suitability of instruments and response rates, (3) adherence to MMD, and (4) sensitivity to change in outcome measurements.

### 3.1. Participant Recruitment and Baseline Characteristics

The enrollment of participants was conducted between June and November 2019 (Figure 3). The recruitment rate from the eligibility assessment to randomization was 41%. After randomization, three people in the IG dropped out due to a combination of the intense physical onsite intervention and lack of time. One person dropped out of the CG without a reason. The total retention rate was 85.2% (IG = 77.7%, CG = 93.7%). Table 2 summarizes the baseline demographics, overall risk for stroke, and cardiometabolic and lifestyle risk factors among the participants. 

The only differences at baseline between the groups were in perceived skill using mHealth technology, which was higher in the CG (CG = 100%, *n* = 15; IG = 64%, *n* = 9; *p* < 0.05), and in diastolic blood pressure (DBP), where the IG had a higher mean DBP (IG = 93 mmHg, CG = 85 mmHg, *p* = 0.044).

### 3.2. Acceptability and Suitability of Instruments and Response Rates

The participants reported no difficulties with the paper or online surveys. The overall response rate of the utilized instruments was 98% at baseline (*n* = 27), 81% (*n* = 24) at the first follow-up, and 75% (*n* = 22) at the 12-month follow-up, with no statistically significant differences between the groups. However, the modified PPR profile (3-day activity diary; see Section 2.7.5) had an overall lower response rate at all three assessments: 71% at baseline (*n* = 24: IG = 12, CG = 12), 44% at the first follow-up (*n* = 16: IG = 8, CG = 8), and 18% at the 12-month follow-up (*n* = 17: IG = 10, CG = 7). The study participants also described the PPR profile as extensive in both the paper and online format.

### 3.3. Adherence

There was a high adherence rate (76%), but it was lower in the second study site (Site 1 = 83%, Site 2= 69%), partly due to more illness absence and one session being scheduled right before a public holiday. Due to COVID-19, follow-up procedures and measurements were affected. To collect weight and blood pressure, some participants used their own devices (weight scale: IG = 4, CG = 6, blood pressure equipment IG = 3, CG = 3); missing data were seen in the 12-month physical performance test (6MWT); and three participants (IG: *n =* 2, CG: *n* = 1) in the 12-month follow-up only participated partially because they were uncomfortable with personal interaction due to COVID-19.

### 3.4. Outcome Measurements

Several outcome measurements were sensitive and able to detect changes at some point within or between the groups at the first follow-up after MMD had ended and at 12 months after baseline).

#### 3.4.1. Overall Stroke Risk

A stroke risk reduction was detected in the IG at follow-up, OR (CI): 7.22 (1.066–48.639) *p* = 0.043, and at 12 months OR (CI): 7.9 (1.221 to >100) *p* = 0.033, compared with the CG. When controlling for covariates (i.e., age, gender, country of birth, living situation, education, income, and work situation), the OR trend was not affected (*p* > 0.05). Table 3 descriptively presents the IG and CG changes in overall stroke risk over time.

#### 3.4.2. Individual Cardiometabolic and Lifestyle Risk Factors

Within the IG, there was a reduction in diastolic blood pressure (DBP) from baseline to the first follow-up, mean difference (SD) 9.6 (12.3) *p* = 0.035, and then to the 12-month follow-up, mean difference (SD) 10.1 (10.6) *p* = 0.007. There was also a clinically meaningful change in both systolic blood pressure (SBP) and DBP within the IG between baseline and the two follow-ups (Table 4). No statistically significant differences in SBP or DBP were detected between the IG and CG at any of the follow-up time points.

At the 12-month follow-up, the IG presented a lower mean weight in absolute numbers (−3.27 kg), whereas the CG presented a higher weight in absolute numbers (+2.9 kg); however, this change was not statistically significant between the IG and CG. Regarding lifestyle habits, within the CG, there was a decreased snack consumption at the first follow-up, median (IQR): −1.0 (1.0) *p* = 0.008, and a higher fruit consumption was detected in the IG in comparison with the CG at the first follow-up, median (IQR): IG 3.0 (2.0), CG 2.5 (1.0) *p* = 0.042.

#### 3.4.3. Stroke Risk Literacy

At the first follow-up, there was a difference within the CG regarding knowledge of risk factors for stroke (*p* = 0.019), and a similar difference within the IG at the 12-month follow-up (*p* = 0.048). At the first follow-up, the IG reported knowledge of more risk factors for stroke than the CG did, median (IQR): IG: 5 (3), CG: 4 (1) *p* = 0.015.

#### 3.4.4. Activity Performance

Both groups changed their activity performance, as measured by COPM [24], from baseline to both follow-ups (Table 5). A clinically meaningful change in activity performance was also detected for the IG between baseline and first follow-up (Table 4). There were, however, no statistically significant changes between the groups at any follow-up. The satisfaction component in the COPM did not reveal any differences between the groups.

#### 3.4.5. Time Use and Experience of Health in Everyday Activities

Between the baseline and first follow-up, the only significant difference within the groups was in time spent on sleep, which increased by 1.08 h in the IG, median (IQR): 1.08 (0.77) *p* = 0.034. However, there were no significant differences between the IG and CG in time spent on sleep at any follow-up. Regarding the participants’ experiences of health in everyday activities, there were no significant differences within the groups. Nevertheless, statistically significant changes was detected in the IG participants’ experience of health in work, median (IQR): IG: 5.0 (1.1) CG: 3.0 (1.5) *p* = 0.005, and household activities related to living independently in the community (I-ADL), median (IQR): IG: 5.0 (1.0) CG: 4 (2.0) *p* = 0.024, at the first follow-up, and in personal activities in everyday life related to self-care (P-ADL), median (IQR): IG: 5.5 (2.0) CG: 4.3 (1.1) *p* = 0.045, at the 12-month follow-up.

#### 3.4.6. Participation in Health-Promoting Activities

Participation in health-promoting activities was measured using a newly developed instrument, PHPA, in which several items showed sensitivity to change. The IG presented a statistically significant change in participation in meaningful activities from baseline to first follow-up, median (IQR): 1.0 (2.0), *p* = 0.041. Both IG and CG increased participation in activities providing an outlet for interests from baseline to first follow-up, median (IQR): IG: 1.0 (1.0) *p* = 0.004, CG 1.0 (1.3) *p* = 0.038. Between the groups, at the first follow-up, there was a statistically significant increase in favor of the IG, in participation in activities that give joy and pleasure, median (IQR): IG: 4.0 (1.0) CG: 3.0 (2.0) *p* = 0.026, and in activities that provide an outlet for interests, median (IQR): IG: 4.0 (1.0) CG: 2.5 (1.0) *p* = 0.039.

#### 3.4.7. Activity Balance

No statistically significant changes were detected within or between the groups in activity balance in the OBQ [36].

#### 3.4.8. Quality of Life

There were no statistically significant changes within or between the groups at the first or 12-month follow-up. A clinically meaningful change in quality of life from baseline to first follow-up was detected within the IG (Table 4).

#### 3.4.9. Physical Performance

There were extensive missing data for the physical performance measurement in both groups at the first follow-up (IG: *n* = 6; CG: *n* = 2) and 12-month follow-up (IG: *n* = 8; CG: *n* = 8) due to COVID-19 restrictions during the ongoing pandemic, resulting in no performed analysis of potential differences. Descriptively, the mean walking distances for both IG and CG were, however, within average (400–700 m) [34] at baseline, first follow-up, and 12-month follow-up.

## 4. Discussion

This pilot study aimed to evaluate the feasibility of the stroke prevention program MMD for persons with a moderate-to-high risk for stroke in a Swedish primary healthcare setting. The results suggest that MMD is feasible. Recruitment was on time, the randomization process was appropriate, and most participants who enrolled in the study completed it and adhered to MMD. The limited number of dropouts and high response rates for most of the instruments indicate that MMD is acceptable. The results also corroborate the acceptability of MMD and its positive progression in terms of lifestyle goals, as indicated in a previous case study [25]. The overall stroke risk reduction observed in the IG also demonstrates the potential usefulness of EEAs in achieving behavioral change, which may be transferable to the prevention of other CVDs and NCDs, also caused by modifiable risk factors [8,47]. With this as a background, a full-scale RCT is considered suitable; however, there are several recommendations for improvements in a full-scale trial and for implementation.

Recruiting participants in clinical trials can be challenging [48,49]. To minimize selection bias, recruitment in the pilot study was carried out through different media and by utilizing primary healthcare clinics situated in geographically and socioeconomically different areas. Nevertheless, as is common in research, willingness to participate was still higher among women and people with higher education. It is therefore important in a full-scale trial to consider additional strategies to recruit persons at risk for stroke in a way that better represents the study population. It may be helpful to examine studies that have successfully recruited hard-to-reach populations, through strategies such as using a flexible and continuously adapting recruitment strategy, combining methods such as using different and relevant forms of social media (e.g., Facebook), involving socio-culturally relevant gatekeepers in the community and/or in healthcare settings [50,51,52], and using respondent-driven sampling [53].

Moreover, relevant data collection procedures and measurements are important in clinical trials. A strength of this study was that it studied both objective and subjective measures using an RCT design. Most participants filled out the comprehensive number of questionnaires and participated in monitoring their blood pressure and weight at three time points, showing that the data collection procedures were acceptable and manageable. Although many studies use them, self-reported measures for lifestyle habits have limitations, such as self-report bias, for example, over-reporting and under-reporting of physical activity [54]. For the full-scale trial, therefore, we recommend combining self-reporting with objective measurements of physical activity using accelerometers, which have been reported as an acceptable and feasible way of measuring physical activity [55].

Regarding the outcome measurements, there is a need for well-designed interventions exploring the usefulness of different instruments measuring stroke risk, activity patterns, and participation and experiences in prevention and health promotion activities. A strength of this study was that it evaluated the use of COPM [24] in a setting the tool had not previously been used in before. COPM aligned well with supporting individual health promotion and was able to detect both statistically significant and clinically meaningful changes in activity performance in the IG and CG. Thus, COPM is likely to be useful in future studies and in the implementation of lifestyle interventions in primary healthcare.

The analysis of stroke risk showed that the IG experienced a greater decrease in overall stroke risk. When controlling for covariates, there was no detected statistical significance, indicating that the stroke risk reduction in the IG remained stable across the analysis; this is a strength of this study, as it shows robustness, albeit with a small sample. It should, however, be noted that the 2018 SRSC [33] was not validity- or reliability-tested, which is a limitation. The instrument also lacked important modifiable risk factors (i.e., a diet with high saturated fat, trans fat, sweetened beverages, salt, and excess calories, and having elevated blood sugar) that were added in a new score card in 2023 [56]. Another instrument, the Stroke Riskometer [57], has also been developed and can be considered for the full-scale trial.

A modified version of the PPR profile [35] was also tested in the present study, in a context within which it has not previously been explored; however, here there was a limitation with the incomplete set of data. In addition, the participants described the instrument as extensive to fill out. One strategy to decrease missing data and increase acceptability could be ecological momentary assessments [58], in which parts of the modified PPR profile are sent out during the day, allowing participants to report their time use and score their experiences in the moment, also limiting recall bias. Other recommendations regarding outcome measures for the full-scale trial include considering the use of outcome measures that showed sensitivity to change in this pilot study (e.g., COPM); eliminating evaluations that are not suitable or that overlap in terms of the aim of the measurement (e.g., quality of life dimensions); and performing psychometric testing of the new PHPA instrument, which added dimensions regarding engagement in activities that can be important in relation to sustainable lifestyle changes, such as experiences of health, meaning, joy, and pleasure in everyday activities [6,23,25,32].

Other important aspects to consider when interpreting the results of this study and planning a full-scale trial include using online or paper surveys, not both; having the same assessors perform the data collection; managing changes in the data collection context during the study period; and striving for similar composition of the IG and CG in terms of stroke risk. In the present study, the IG had more people with high stroke risk than the CG in this study. Although the difference was not statistically significant, it may have affected the results in terms of stroke risk reduction. Moreover, the attendance at one of the MMD prevention program groups was lower than our prespecified goal, which may have affected the results. Measures should be taken in the full-scale RCT to improve adherence (e.g., not scheduling group sessions close to public holidays).

## 5. Conclusions

This pilot study provides useful information on important aspects of evaluating the stroke prevention program MMD. The results demonstrate the potential usefulness of EEAs in achieving behavior change, which may be transferable to the prevention of other CVDs and NCDs, also caused by modifiable risk factors. Based on the information gained from the results of this study, which show reduced stroke risk in the IG, as well as acceptability of MMD and positive progression in terms of lifestyle goals, a full-scale RCT including an evaluation of the effectiveness of MMD is suggested. A full-scale trial will, however, require the following improvements: the recruitment of a relevant sample, the use of relevant and reliable outcome measures, and the implementation of strategies to limit missing data.

## Figures and Tables

**Figure 1 ijerph-20-06828-f001:**
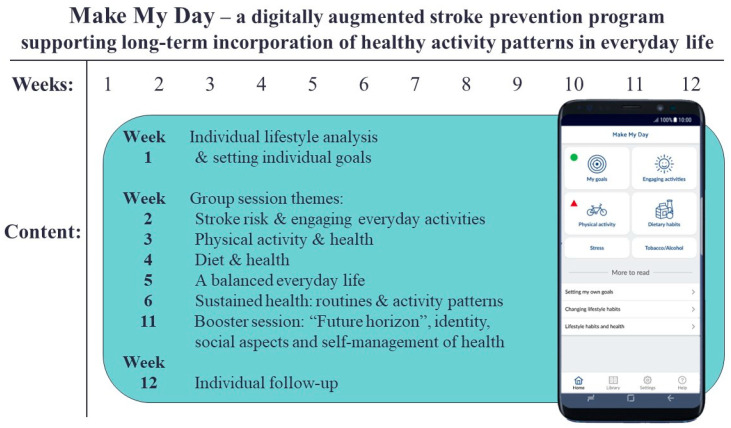
Outline of prevention program Make My Day.

**Figure 2 ijerph-20-06828-f002:**
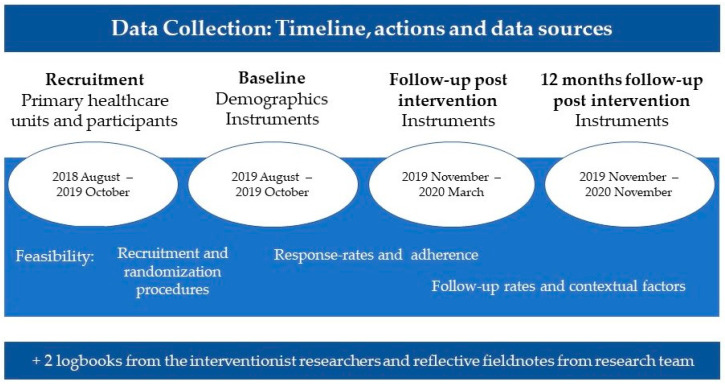
Overview of the timeline, actions, and data sources.

**Figure 3 ijerph-20-06828-f003:**
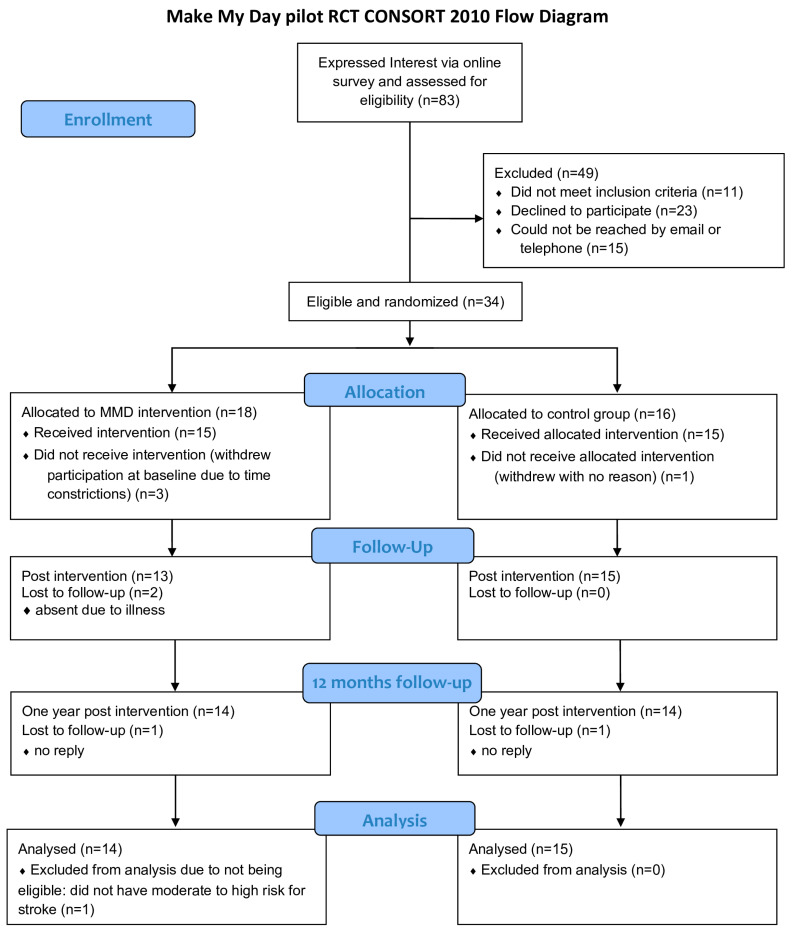
Flow chart of MMD pilot RCT.

**Table 1 ijerph-20-06828-t001:** Instruments used for data collection.

Outcome	Instrument	Purpose	Measure
Stroke risk	Stroke Risk Score Card (SRSC) [33]	Overall stroke risk combining modifiable and non-modifiable risk factors	8 domains with three categories each. Total score, summing up each category: ≥3 = High Stroke Risk, 4–6 = Caution, 6–8 = Low Stroke Risk
Anthropometric measures	Body Mass Index (BMI)	Indicator for body fat and weight status	<18.5 = Underweight, 18.5–24.9 = Normal, 25.0–29.9 = Overweight, >30.0 = Obesity
	Blood pressure (BP) [40,41]	Measurement of systolic and diastolic blood pressure	High blood pressure: >140/90, Elevated blood pressure: 120–139/80–89, Normal blood pressure: <120/80. A decrease of 10 mm Hg in systolic blood pressure and 5 mm Hg in diastolic blood pressure was considered a clinically meaningful change.
Lifestyle habits	Lifestyle habit survey [42].	Estimation of lifestyle habits in everyday life: tobacco use, alcohol consumption, physical activity, and eating habits.	Four habits with 11 questions in total, subdivided into 4–5 levels of performance during a week or the last months. For example, “how often do you eat fruit or berries?” 1 = Two times each day or more 2 = One’s a day 3 = A couple of times during a week 4 = One time, or less, during a week.
Physical performance	6 min walk test (6MWT) [34]	Physical performance while walking 6 min.	Measurements before and after 6 min walk between a marking of 30 m: distance in meters, saturation (SpO_2_), pulse (beat/minute), blood pressure (mm Hg systolic and diastolic), and estimating shortness of breath and leg fatigue with Borg Scale (1–10).
Activity performance and satisfaction	Canadian Occupational Performance Measure (COPM) [24].	Perceived performance and satisfaction with activities in everyday life.	Scale ranging from 1 to 10 in two aspects: (i) current performance, 1 = not able to perform the activity at all to 10 = able to do it extremely well, and (ii) satisfaction with doing, 1 = not satisfied to 10 = extremely satisfied.
Activity patterns	The Daily Experiences of Pleasure, Productivity and Restoration Profile (PPRP) [35].	Perceived pleasure, productivity, and restoration during three days (two week days and one weekend day).	Time as measured in hours. Scale of 1–7 in four dimensions (pleasure, productivity, restoration and health) related to the time use. For example: 1 = Extreme displeasure 2 = Moderate displeasure 3 = A little displeasure 4 = Neither pleasure nor displeasure 5 = Moderate pleasure 6 = Extreme pleasure.
Participation in health-promoting activities	PHPA questionnaire	Perceived participation in activities in everyday life that can increase or decrease health.	10 statements with 5 levels of agreement: 1 = No, do not participate, 2: Sometimes (less than once a week), 3 = Often (up to twice a week), 4 = regularly (three or more times a week), 5 = Daily.
Perception of balance in everyday life	Occupational Balance Questionnaire (OBQ) [36]	Perception of balance within or between different occupations in everyday life.	13 statements with 4 level of agreements: 0 = Do not agree at all 1 = Agree partially 2 = Agree a lot 3 = Totally agree, with a sum ranging from 0–39.
Stroke risk literacy	Items from a stroke risk knowledge and awareness of stroke risk questionnaire [37]	Three questions indicating literacy of stroke risks, stroke risks that can be impacted by modifiable means, and scoring of one’s one stroke risk.	Counting of stroke risk factors were counted from 1–7, and the scoring of own risk by 1–10.
Quality of life	EQ-5D-3L Questionnaire [38]	General life quality. Perceived state of health in five aspects: mobility, hygiene, main activities, pain, and anxiety.	Index scale from 0 to 1, based on scores from 1 to 3, subdivided into three levels of severity: 1 = no problem, 2 = some/moderate problems, 3 = extreme problems.
	EQ-Visual Analogue [38] Scale (EQ-VAS)	Perceived state of health.	Scale 0–100: 0 = worst possible health, 100 = best possible health.
Life Satisfaction	Life Satisfaction Scale11 (LiSat-11) [39]	Perceived satisfaction with life	Scale 1–6: 1 = not satisfied to 6 = very satisfied

**Table 2 ijerph-20-06828-t002:** Demographics and stroke risk characteristics of the study participants at baseline.

Variables	Intervention Group *n* = 14	Control Group *n* = 15
**Age, *Mean (SD)***	61.9 (8.5)	59.7 (7.5)
**Education, *Mean (SD)***	13.5 (1.8)	13.7 (1.9)
**Sex, Female**	8 (57)	12 (80)
**Country of birth country, Sweden**	9 (64)	11 (73)
**Living situation** Alone	8 (57)	7 (47)
Together	6 (43)	8 (53)
**Area of housing/living** ^a^ Good socioeconomic conditions	11 (79)	13 (87)
Socioeconomic challenges	3 (21)	2 (13)
**Work status** Working	6 (43)	8 (53)
Not working ^b^	8 (57)	7 (47)
**Yearly income, in euro** >58,000	3 (21)	1 (7)
<58,000 >19,300	10 (72)	11 (73)
<19,300	1 (7)	3 (20)
**mHealth technology use** Interest	13 (93)	15 (100)
Skills ^c^	9 (64)	15 (100)
**Overall stroke risk** ^d^ High risk	9 (64)	6 (40)
Caution	5 (36)	9 (60)
**Modifiable risk factors** Atrial fibrillation ^e^	1 (7)	1 (7)
Diabetes type 2 or borderline	5 (36)	6 (40)
High or elevated blood pressure	11 (79)	9 (60)
Smoking	4 (29)	0 (0)
Overweight ^f^	7 (50)	9 (60)
Insufficient physical exercise ^g^	10 (71)	14 (93)
Insufficient physical activity ^h^	7 (50)	9 (60)
Insufficient vegetable consumption ^i^	5 (36)	8 (53)
Insufficient fruit and berry consumption ^i^	4 (29)	8 (53)
Limited weekly snack consumption ^j^	0 (0)	3 (20)
Recurring stress in everyday life ^k^	10 (71)	12 (80)

^a^ According to the Swedish National Board of Housing, Building and Planning [43] (https://segregationsbarometern.boverket.se/labbet/#/omradesstatistik/map?indicator=0-1,2,3,4,5&location=riket&bg=0) (accessed on 13 March 2023). ^b^ Not working meant being either retired, unemployed, or on sick leave. To not risk identification of any participants, counts of each subgroup are not presented. ^c^ A statistically significant difference was observed between Intervention group and Control group (*p* = 0.017) regarding mHealth skills, based on two self-reported questions about the use of mHealth. ^d^ According to a stroke risk score [33]. ^e^ Diagnosed and receiving medication. ^f^ BMI >25. ^g^ Moderate-to-intense physical exercise (<150 min moderate or 75 min intensive weekly) [44]. ^h^ Physical activity (<300 min weekly) [44]. ^i^ <1–2 portions a day [45]. ^j^ <1–3 portions each week [45]. ^k^ Psychosocial stress as risk factors for stroke [46].

**Table 3 ijerph-20-06828-t003:** Participants’ overall stroke risk at baseline, follow-up, and 12 months presented in numbers and percentage.

	Stroke Risk	Intervention Group *n* = 14	Control Group *n* = 15
Baseline	High risk	9	6
	Caution	5	9
Follow up	High risk	5 ^b^	5 ^a^
	Caution	4 ^a^	8 ^a^
	Low risk	2	1
12 months	High risk	5 ^a^	9
	Caution	7 ^a^	4 ^a^
	Low risk	0	1

Missing value: number of missing participants, ^a^ = 1 person, ^b^ = 2 persons.

**Table 4 ijerph-20-06828-t004:** Clinically meaningful changes from baseline to follow-up, and from baseline to 12 months, within each group.

Measures	Baseline	1st Follow up	12-Month Follow-up	Mean Difference from 1st Follow-up	Mean Difference from 2nd Follow-up	Clinical Cut off Score
Intervention group						
SBP ↓	141	130	131	**−11 ***	**−10 ***	10
DBP ↓	93	85	85	**8**	**8**	5
COPM (1–10) ↑	4.1	6.6	5.8	**2.6 ***	1.7	>2
EQ-VAS (1–100) ↑	65	75	74	**10 ***	9	>10
Control group						
SBP ↓	131	129	127	−2	−3	10
DBP ↓	85	87	83	+2	−2	5
COPM (1–10) ↑	3.8	5.4	5.8	1.6	1.9	>2
EQ-VAS (1–100) ↑	63	65	72	2	9	>10

SBP Systolic blood pressure, DBP Diastolic blood pressure, COPM Canadian Occupational Performance Measure [24], EQ-VAS (EQ-Visual Analogue Scale) [38]. Clinically meaningful cut-off score for each instrument: Systolic/Diastolic blood pressure—A decrease of 10 mm Hg in systolic blood pressure and 5 mm Hg in diastolic blood pressure is considered a clinically meaningful change; COPM—an increase of ≥2 is considered a clinically meaningful change; EQ-VAS—a change of ≥10 indicates a clinically meaningful change. ↑ a higher score indicates a better outcome, ↓ a lower score indicates a better outcome. * A clinically meaningful change within the group between follow-ups was detected.

**Table 5 ijerph-20-06828-t005:** Differences from baseline to follow-up, and from baseline to 12 months, within each group.

Measures	Intervention Group				Control Group			
	Baseline to Follow-up		Baseline to 12 Months		Baseline to Follow-up		Baseline to 12 Months	
	*n* = 14	*p*-Value	*n* = 14	*p*-Value	*n* = 15	*p*-Value	*n* = 15	*p*-Value
COPM (1–10) *Median (IQR)*	2.5 (2.7) ^b^	**0.050** *	1.8 (2.3) ^a^	**0.008** *	1.0 (2.5) ^b^	**0.002** *	1.6 (3.8) ^a^	**0.004** *

COPM Canadian Occupational Performance Measure. Missing value: number of missing participants indicated with ^a^ = 1 person, ^b^ = 2 persons * Significant level *p* < 0.05.

## Data Availability

The data used in this study contain sensitive information about the study participants. The current approval by the Swedish Ethical Review Authority does not include data sharing. A minimal dataset could be shared on request from a qualified investigator for the purpose of replicating the present study, with data transfer in agreement with EU legislation on the general data protection regulation and approval by the Swedish Ethical Review Authority.

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
