# Peer review of "The Feasibility of Make My Day—A Randomized Controlled Pilot Trial of a Stroke Prevention Program in Primary Healthcare"

_ijerph, 2023, doi:10.3390/ijerph20196828_

Round 1

Reviewer 1 Report

This pilot study suggests that the MMD stroke-prevention program shows promise in reducing stroke risk and achieving positive lifestyle goals. This is meaningful research but so many details need to double check:

1.      Please correct the typo of “(F)igure 3)” in line 349.

2.      In the Flow chart, why does the 12-month follow-up has even one person more than follow-up participants (n=14 vs n=13) in the MMD group? Does the final number in the analysis involve the 12-month lost follow-up participant? Or where the one not eligible person came from in the MMD group and how did the number more than 12-month number in the control group (n=15 vs. n=14)?

3.      Since the pilot study has a small sample size. Did the author use Fisher’s exact test to adjust the analysis result? If yes please also list in the data analysis section.

4.      Please try to re-format Table 2 to make it easier to read for the first column, especially for those with multiple levels of variables.

5.      In Table 2, please double check the footnote b, the number of not working in the footnote is not the same as in the table (n=14: 9+2+3 vs. n=15: 8+7).

6.      For mHealth technology use, is that a question people can choose both interest and skills?

7.      The extension of the abbreviation of SBP/DBP for Table 4, 5, and 6 are incorrect.

8.      For Table 5, does the * mean there were meaningful changes? If yes please also explain in the footnote.

9.      In line 422, the fruit consumption difference between IG and CG groups was shown in Table 6, not 5.

Author Response

Dear reviewers and editor, 

Thank you for reading our manuscript in such detail and providing these comments and opportunity for us to revise our manuscript. The comments have provided guidance for us to make our results and main messages more clear. We are grateful for all you time spent to help us make this manuscript better. 

Sincerly Emelie and co-authors

Reviewer 2 Report

The major goal of this pilot study is to find the acceptance and suitability of MMD for stroke prevention program. MMD may be helpful in reducing the stroke risk and may encourage the people to find a suitable way to decrease the stroke risk. However, it is difficult to reach at any meaningful conclusion with such a small sample size and many limitations due to pandemic. There are few things that needs to improvise.

1.  The manuscript is very descriptive, and the readers will lose interest to get the message out for example: Line 384-397 can be explained in few lines.

2. The outcome measures are not focused and seems complicated. In case it categorized into two modifiable and non-modifiable risk factors and most of the risk factors comes under this category. 

3. Table 5 is showing clinically meaningful changes, then there is no need for a long table 4.

4. As authors agreed that they are many limitations to this study, and they need to work on it before extending the study.

5. This outcome needs to publish as a report rather than manuscript.

Author Response

(The authors gave the same response as above.)

Reviewer 3 Report

Scientific literature supports the notion that every movement counts and that integrating everyday activities (EEAs) into daily life can be an effective strategy for stroke prevention and overall health improvement. For a comprehensive stroke prevention strategy, a combination of EEAs, purposeful exercise, and other healthy lifestyle habits is recommended.

However, I may suggest some points, which may be improved:

A/It is a small-scale pilot study, a limitation acknowledged by the authors themselves. The conclusions drawn are already well established in existing evidence and could potentially be extrapolated to cardiovascular risk in general, beyond just stroke patients;

B/the baseline cardiovascular risk of both groups is not described. This missing information could provide context for interpreting the study's findings.

C/the tables, particularly Table 4 and Table 6, contain dense content and could benefit from redesigning to enhance their visualization and comprehension.

Author Response

(The authors gave the same response as above.)

Round 2

Reviewer 3 Report

The article has clear potential in addressing stroke, but I believe that the tables (especially tables 4 and 5) are still too complex to capture the reader's interest. It's an article in which it contrasts the scarcity of cases with the excess of statistical data.

Author Response

Dear reviewer, thank you for the throrough review and support in improving the manuscript. Attached there are detailed information about the revisions.

All the best, Emelie and co-authors